# B4GAT1 is the priming enzyme for the LARGE-dependent functional glycosylation of α-dystroglycan

Jeremy L Praissman[1,2], David H Live[1], Shuo Wang[1], Annapoorani Ramiah[1], Zoeisha S Chinoy[1], Geert-Jan Boons[1], Kelley W Moremen[1,2], Lance Wells[1,2]*

[1]Complex Carbohydrate Research Center, University of Georgia, Athens, United States; [2]Department of Biochemistry and Molecular Biology, University of Georgia, Athens, United States

**Abstract** Recent studies demonstrated that mutations in B3GNT1, an enzyme proposed to be involved in poly-N-acetyllactosamine synthesis, were causal for congenital muscular dystrophy with hypoglycosylation of α-dystroglycan (secondary dystroglycanopathies). Since defects in the O-mannosylation protein glycosylation pathway are primarily responsible for dystroglycanopathies and with no established O-mannose initiated structures containing a β3 linked GlcNAc known, we biochemically interrogated this human enzyme. Here we report this enzyme is not a β-1,3-N-acetylglucosaminyltransferase with catalytic activity towards β-galactose but rather a β-1,4-glucuronyltransferase, designated B4GAT1, towards both α- and β-anomers of xylose. The dual-activity LARGE enzyme is capable of extending products of B4GAT1 and we provide experimental evidence that B4GAT1 is the priming enzyme for LARGE. Our results further define the functional O-mannosylated glycan structure and indicate that B4GAT1 is involved in the initiation of the LARGE-dependent repeating disaccharide that is necessary for extracellular matrix protein binding to O-mannosylated α-dystroglycan that is lacking in secondary dystroglycanopathies.

*For correspondence: lwells@ccrc.uga.edu

**Competing interests:** The authors declare that no competing interests exist.

**Reviewing editor**: Suzanne R Pfeffer, Stanford University, United States

## Introduction

Glycosylation is the most abundant and diverse post-translational modification of proteins (*Varki, 2011*). The synthesis of complex glycans is catalyzed by the action of over 200 individual glycosyltransferases in humans (*Moremen et al., 2012*). For most of the enzymes studied to date, there is exceptional selectivity for the donor sugar nucleotide and the underlying acceptor glycan as well as stereo- and linkage-specificity for the catalyzed additions (*Moremen et al., 2012*). Mutations in the genes encoding many of these glycosyltransferases have been established as causal for a variety of human diseases including congenital muscular dystrophy, specifically the secondary dystroglycanopathies (*Barresi and Campbell, 2006*; *Freeze, 2007*; *Wells, 2013*; *Praissman and Wells, 2014*).

At least a dozen enzymes/proteins are involved in the synthesis of O-mannose-initiated glycans that when defective lead to various forms of congenital muscular dystrophy, termed secondary dystrogly-canopathies, that range from the phenotypically mild Limb-Girdle to the severe Walker-Warburg muscular dystrophies (*Muntoni et al., 2011*; *Mercuri and Muntoni, 2012*; *Dobson et al., 2013*; *Praissman and Wells, 2014*). Until recently (*Lommel et al., 2013*; *Vester-Christensen et al., 2013*; *Winterhalter et al., 2013*), α-dystroglycan, a central component of the dystrophin-glycoprotein complex that serves to connect the cytoskeleton inside the cell with the extracellular matrix outside the cell, was the only well established O-mannoslyated mammalian protein (*Barresi and Campbell, 2006*; *Endo and Manya, 2006*). The proper O-mannosylation of α-dystroglycan is essential for its ability to bind to components of the extracellular matrix including laminin (*Ervasti and Campbell, 1993*; *Barresi and Campbell, 2006*).

**eLife digest** Dystroglycan is a protein that is essential for muscles to function correctly, and helps to connect the interior framework of muscle cells with the external matrix of molecules that hold the cells together in the tissue. As is the case for many proteins, dystroglycan must have particular carbohydrate molecules joined to it in order to work correctly. Enzymes called glycosyltransferases assist with the reactions that build the carbohydrates on a protein.

Mutations in multiple glycosyltransferases that add carbohydrates to dystroglycan can cause a group of diseases that are characterized by a progressive loss of muscle function, known as congenital muscular dystrophies. Praissman et al. use biochemical experimentation to investigate the role of one of these enzymes, known as B3GNT1. The enzyme's name is based on a code that describes which carbohydrate it helps to bind to proteins. However, Praissman et al. (and independently, Willer et al.) discovered that this enzyme actually works with a different donor and acceptor than previously thought, and so should be called B4GAT1 instead.

Praissman et al. propose that the B4GAT1 enzyme starts the process of forming the carbohydrate structures that help muscle cells bind to the muscle tissue matrix. B4GAT1 forms short carbohydrates on the surface of the part of dystroglycan that sits on the surface of cells. These carbohydrates are then extended into longer chains by another glycosyltransferase called LARGE. The results of Praissman et al. suggest that another enzyme is also involved in this process, which will require further studies to identify. Understanding the role of B4GAT1 and other glycosyltransferases that build functionally glycosylated dystroglycan could help to develop treatments for diseases such as muscular dystrophies.

In recent years, significant progress has been made in defining the multitude of O-mannose structures produced in multicellular animals including partial elucidation of the LARGE-dependent functional glycan structure (*Stalnaker et al., 2010*; *Yoshida-Moriguchi et al., 2010*; *Hara et al., 2011*; *Stalnaker et al., 2011a*; *Stalnaker et al., 2011b*; *Harrison et al., 2012*; *Inamori et al., 2012, 2013*; *Live et al., 2013*; *Panin et al., 2014*). Recently, reports in the literature have connected mutations in B3GNT1 with congenital muscular dystrophies (*Buysse et al., 2013*; *Czeschik et al., 2013*; *Shaheen et al., 2013*) and specifically to the defective glycosylation of α-dystroglycan (*Jae et al., 2013*) even though no O-mannose glycan structures containing a β3-GlcNAc have yet to be elucidated (*Praissman and Wells, 2014*).

In 1997, B3GNT1 (also known as iGnT) was reported to be the first successfully cloned β-1,3-N-acetylglucosaminyltransferase involved in the synthesis of poly-N-acetyllactosamine (*Sasaki et al., 1997*). Poly-N-acetyllactosamine chains consist of the repeating disaccharide -β3-GlcNAc-β4-Gal-, a structure termed the i-antigen when unbranched (*Ujita et al., 2000*; *Zhou, 2003*). Poly-N-acetyllactosamine is a prevalent glycan substructure found in N-glycans and O-glycans on proteins whose abnormal levels have been associated with human diseases including cancer (*Ujita et al., 2000*; *Zhou, 2003*; *Ho et al., 2013*; *Lu et al., 2014*). The Fukuda group used an expression cloning strategy enriching for plasmids containing cDNA inserts that substantially increased poly-N-acetyllactosamine, as judged by antibody binding, on the surfaces of Namalwa KJM-1 cells (*Sasaki et al., 1997*). Shortly thereafter, work carried out by Hennet's and Sasaki's groups led to the cloning of three other β-1,3-N-acetylglucosaminyltransferases, B3GNT2, 3, and 4 (*Zhou et al., 1999*; *Shiraishi et al., 2001*). However, papers as recently as 2008 continued to use conflicting nomenclature for B3GNT2 by calling it B3GNT1 causing some confusion within the field (*Biellmann et al., 2008*). Importantly, B3GNT2, B3GNT3 and B3GNT4 were found to share motifs with β3-galactosyl-transferases as well β3-GalNAc-transferases (*Zhou et al., 1999*; *Shiraishi et al., 2001*; *Togayachi et al., 2006*). These three enzymes are in the same Carbohydrate-Active Enzymes database (CAZy) family and group together based on primary sequence similarity and motifs hypothesized to play a role in forming the β3 linkage (*Zhou et al., 1999*; *Shiraishi et al., 2001*; *Togayachi et al., 2006*). B3GNT1 lacks these motifs and is in a different CaZY family, interestingly one shared with LARGE and LARGE2 that contain both xylosyl- and glucoronsyl–transferase activity and which functionally modify O-mannosylated glycans (*Inamori et al., 2012*; *Goddeeris et al., 2013*; *Inamori et al., 2013*). Furthermore, it has been reported that the B3GNT1 enzyme is a binding partner for LARGE in mammalian cells (*Bao et al., 2009*).

The combination of data suggesting substantial divergence of B3GNT1 from other B3GNTs and the similarity in primary sequence and disease association with LARGE led us to reexamine the enzymatic activity of B3GNT1 to resolve the apparent inconsistencies and establish its potential enzymatic role in O-mannosylation and secondary dystroglycanopathies. Here we report that B3GNT1 is in fact B4GAT1 (β-1,4-glucuronyltransferase 1) that generates the substrate for the LARGE-dependent repeating disaccharide that is required for interaction of O-mannosylated α-dystroglycan with extracellular matrix proteins.

## Results

### B3GNT1 (B4GAT1), unlike B3GNT2, is not a N-acetylglucosaminyltransferase

A secreted (lacking the trans-membrane domain) epitope-tagged form of human B3GNT1 was recombinantly expressed in HEK293F cells, purified, and the epitope tag removed before enzymatic characterization. Purity and enzyme identity was assessed by Coomassie G-250 staining of SDS-PAGE separated protein as well as by shotgun proteomics using reverse-phase liquid chromatography-nanospray tandem mass spectrometry (LC-MS/MS) following tryptic digestion (*Figure 1—figure supplement 1*, *Supplementary file 1*). Incubation of the purified protein with UDP-[$^3$H]GlcNAc and an enzymatically synthesized and purified substrate, Gal-β4-GlcNAc-β-pNP (*Figure 1—figure supplement 2*), under previously reported buffering conditions for B3GNTs (*Zhou et al., 1999*; *Shiraishi et al., 2001*) failed to result in a detectable product as measured by incorporation of radioactivity into substrate (*Figure 1*). In order to validate our acceptor sugar and buffering conditions, we expressed, purified, and characterized a secreted epitope-tagged form of human B3GNT2 (*Figure 1—figure supplement 1*, *Supplementary file 1*). Side-by-side extended (overnight) incubations of B3GNT1 and the B3GNT2 enzyme preparations with acceptor and sugar nucleotide showed clear N-acetylglucosaminyltransferase activity toward the acceptor only with B3GNT2, as assayed by radioactive incorporation of GlcNAc into the N-acetyllactosamine acceptor (*Figure 1*, glycans displayed in symbolic representations [*Varki et al., 2009*]).

### B3GNT1, but not B3GNT2, is a glucuronyltransferase that uses xylose as an acceptor

Given that B3GNT1 clustered into CAZy family GT49, the family containing the glucuronyltransferase domain of LARGE, we performed a multiple sequence alignment (*Figure 2A*). B3GNT1 aligned strongly with LARGE and LARGE2 in the glucuronyltransferase domain DXD-motif region typically involved in metal ion dependent sugar nucleotide binding. This DXD-motif was shown to be necessary for the glucuronyltransferase activity of LARGE and LARGE2 consistent with probable sugar nucleotide binding in this region (*Inamori et al., 2013*). B3GNT2 did not align well with B3GNT1 or the LARGE proteins (*Figure 2A*). We hypothesized that B3GNT1 might have glucuronyltransferase activity contrary to its proposed enzymatic function and thus carried out a screen using various tagged monosaccharide acceptors. Incubation of B3GNT1 and UDP-GlcA with multiple glycan acceptors resulted in significant transfer only to α- and β-xylopyranosides, as measured by radioactive transfer from UDP-[$^{14}$C]GlcA (*Figure 2B*). B3GNT2 did not transfer GlcA to either anomeric form of xylopyranoside tested (data not shown).

### B3GNT1, unlike LARGE, is not selective for the stereochemistry of the acceptor xylose

We next expressed, purified and characterized a secreted form of human LARGE (*Figure 1—figure supplement 1*, *Supplementary file 1*) whose activity we investigated. Our results confirm previous reports that LARGE transfers GlcA to α-xylopyranosides but not β-xylopyranosides (*Figure 2C*, [*Inamori et al., 2012*, *2013*]). In contrast, B3GNT1 does not appear to be hampered by the stereochemistry of the acceptor sugar (*Figure 2B,C*). We also confirmed the disaccharide products by LC-MS/MS (*Figure 2D–F*).

### Soluble forms of B3GNT1 and LARGE do not form a complex in vitro

In order to investigate whether a complex formed between our soluble enzymes, we incubated the purified forms of the two enzymes together with only the LARGE enzyme being epitope tagged. Following re-purification of LARGE based on the epitope tag, we were unable to detect the presence of B3GNT1 suggesting that the soluble forms of the proteins do not interact in vitro (*Figure 2—figure supplement 1*).

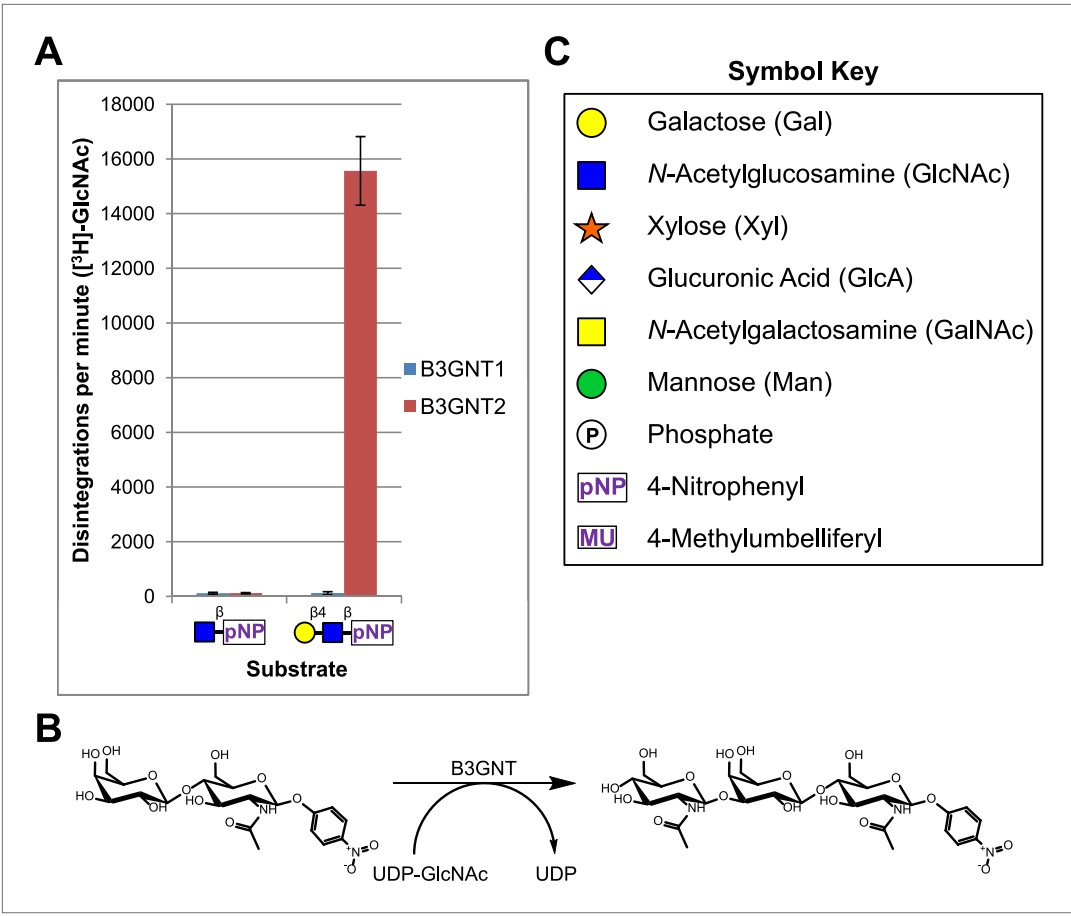

**Figure 1**. B3GNT1 does not possess β-1,3-N-acetylglucosaminyltransferase activity. (**A**) The β-1,3-N-acetylglucosaminyltransferase activity of B3GNT1 towards pNP tagged N-acetyllactosamine was compared to that of B3GNT2 (reaction scheme presented in (**B**)). Incubations were carried out overnight at 37°C in 0.1 M MES pH 6.5 containing 10 mM MnCl2, 40,000 DPM UDP-[³H]GlcNAc and 2 mM non-radioactive UDP-GlcNAc. pNP-sugars were isolated from sugar nucleotide by reverse-phase C18 spin columns. UDP-[³H]GlcNAc transfer to Gal-**β4**-GlcNAc-**β**-pNP was measured by liquid scintillation counting. Averaged results from three independent experiments with error bars indicating standard deviation are shown. GlcNAc-**β**-pNP was used as a negative control acceptor. (**C**) Relevant sugar code symbols from 'Essentials of Glycobiology' as well as other symbols used throughout the paper are shown.

The following figure supplements are available for figure 1:

**Figure supplement 1**. Coomassie brilliant blue G-250 stained SDS-PAGE gels of purified enzyme samples.

**Figure supplement 2**. Gal-β4-GlcNAc-β-pNP synthesized using bovine B4GALT1 LC-MS and LC-MS/MS spectra of Gal-β4-GlcNAc-β-pNP produced using B4GALT1.

## B3GNT1 is more catalytically efficient than LARGE with monosaccharide acceptors

We performed kinetic analysis of B3GNT1 and LARGE using the α-xylopyranoside as an acceptor and determined only minor differences (3.0 and 6.0 mM, respectively) in Km values for the acceptor (*Table 1*). However, at 2 mM UDP-GlcA, B3GNT1 has a specific activity that is 48 times higher than LARGE for α-xylopyranoside as the acceptor (*Table 1*). For B3GNT1, we were also able to use β-xylopyranoside as an acceptor and determined that the enzyme was more than twice as efficient with this acceptor as opposed to its anomer, α-xylopyranoside, though the Km value for the acceptor (4.0 mM) was only slightly altered (*Table 1*). Thus, B3GNT1 possesses a significantly higher specific activity, turnover rate and catalytic efficiency than LARGE toward the monosaccharide α-xylopyranoside acceptor and is even more efficient with the β-anomer that LARGE is not able to utilize (*Table 1*).

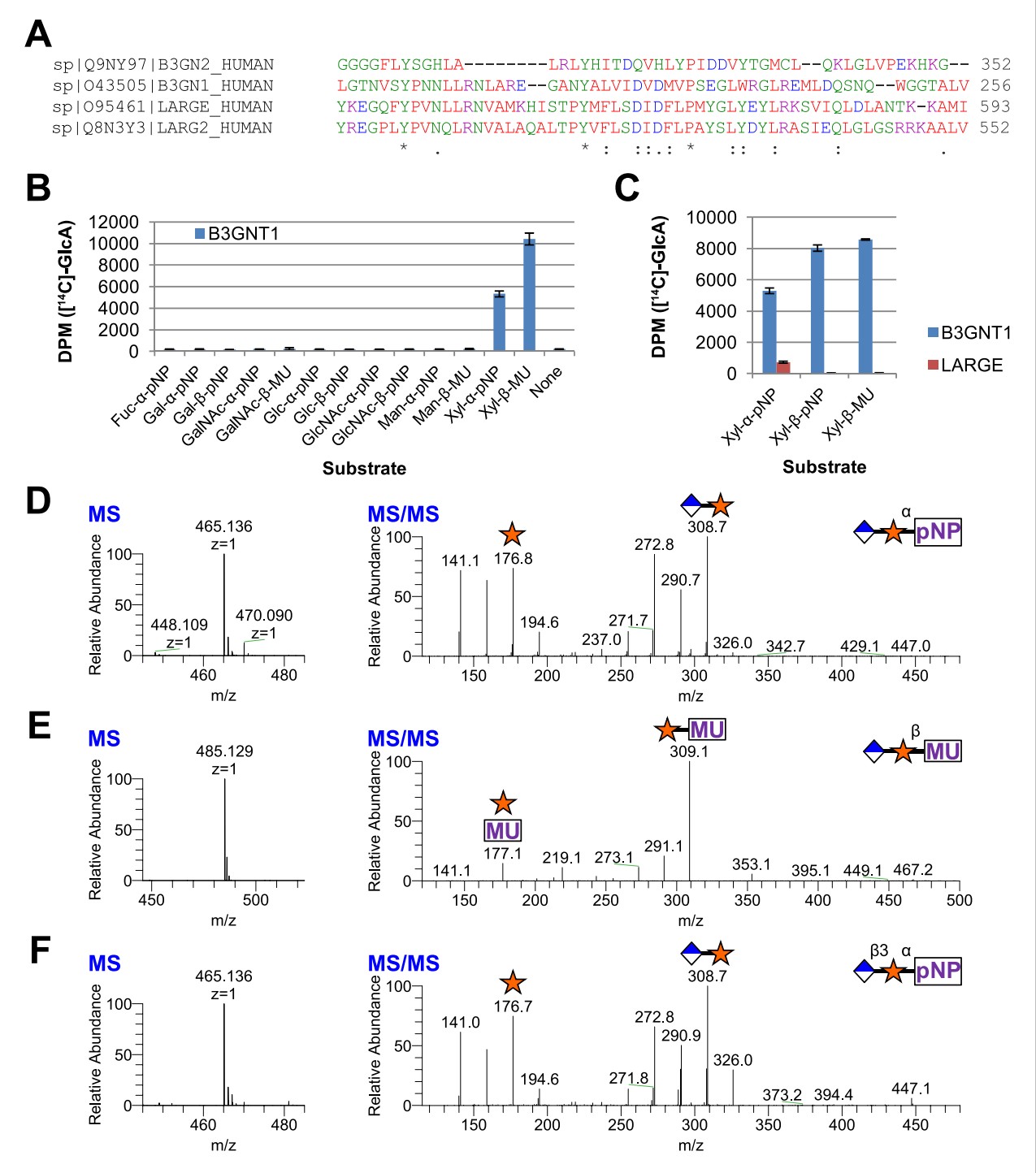

**Figure 2**. B3GNT1 is a glucuronyltransferase that uses xylose as an acceptor. B3GNT1 is in CAZy family GT49 along with LARGE and LARGE2. (**A**) Clustal Omega multiple sequence alignment excerpt showing the strong alignment of B3GNT1 with the glucuronyltransferase domain of LARGE and LARGE2, including the DXD motif shown to be important for activity. B3GNT2 does not align well with the other inputs. (**B**) UDP-[$^{14}$C]GlcA transfer screen assayed by liquid scintillation counting (disintegrations per minute–DPM), averaged results of three independent experiments with error bars indicating standard deviation. B3GNT1 transfers to xylose in both anomeric configurations. (**C**) LARGE transfers only to α-linked xylose as previously reported. (**D**) LC-MS and LC-MS/MS data showing B3GNT1 transfer to Xyl-α-pNP. The ammonium adduct is the dominant species (465.136) however the protonated species is observable at 448.109. The majority of unlabeled fragment peaks represent losses of water from labeled peaks. (**E**) LC-MS and LC-MS/MS data showing

*Figure 2. Continued on next page*

*Figure 2. Continued*

B3GNT1 transfer of UDP-GlcA to Xyl-β-MU. (**F**) LC-MS and LC-MS/MS data showing LARGE transfer to Xyl-α-pNP. Reactions were carried out overnight in 0.1 M MES pH 6.5 containing 10 mM MnCl2, 5 mM MgCl2, 2 mM substrate, 40,000 DPM UDP-[¹⁴C]GlcA and 2 mM non-radioactive donor.
The following figure supplement is available for figure 2:

**Figure supplement 1**. Silver stained SDS-PAGE gel of complex formation assay samples.

## B3GNT1 is a B4GAT unlike LARGE that harbors B3GAT activity

To further characterize the product of B3GNT1 compared to LARGE, we carried out large-scale transfer reactions with the proven substrates of each enzyme. Disaccharide products were purified using reverse phase C18 HPLC and analyzed by multiple NMR-based experiments (*Figure 3*, *Figure 3—figure supplement 1*, *Supplementary file 2*). Glycosidic linkages and anomeric configurations were determined by NMR (*Bock and Pedersen, 1974*; *Van de Ven, 1995*; *Wishart et al., 1995*) that clearly demonstrates that the B3GNT1 and LARGE products share the same stereochemistry but different linkages of the terminal glucuronic acid to the underlying xylose (*Figure 3*, *Figure 3—figure supplement 1*, *Supplementary file 2*). From this analysis, we determined that B3GNT1 is a xylopyranoside β1,4-glucuronyltransferase that we designate, using standard convention, B4GAT1 and, as previously described, we confirmed that LARGE contains xylopyranoside β1,3- glucuronyltransferase activity (*Inamori et al., 2012*).

## The B4GAT1 disaccharide products are substrates for the xylosytransferase activity of LARGE

Since B4GAT1 transfers GlcA to both anomers of xylopyranoside in contrast to LARGE and produces a β1,4-linkage as opposed to a β1,3-linkage, we sought to determine if the xylosyltransferase domain of LARGE would transfer to products of B4GAT1. We tested GlcA-β4-Xyl-α-pNP and GlcA-β4-Xyl-β-MU (products of B4GAT1) as well as GlcA-β3-Xyl-α-pNP (the product of LARGE as a positive control), each of which we produced and then purified by reverse phase C18 HPLC before use. Reactions were carried out overnight at 37°C with UDP-Xyl as the donor. Substrates and products were desalted by reverse phase C18 spin columns and then subjected to C18 reverse-phase HPLC. Distinct substrate and product peaks were detected for each of the three potential acceptors tested and the identity of each peak was confirmed by LC-MS/MS (*Figure 4*). Hence, LARGE can extend both disaccharide products of B4GAT1 and its own disaccharide product with α3-linked xylose.

## Trisaccharides are not efficient substrates for B4GAT1, unlike the glucuronyltransferase activity of LARGE

Having produced three different trisaccharides with non-reducing end terminal xylose, Xyl-α3-GlcA-β4-Xyl-α-pNP, Xyl-α3-GlcA-β4-Xyl-β-MU (sequential products of B4GAT1 and xylosyltransferase activity of LARGE), and Xyl-α3-GlcA-β3-Xyl-α-pNP (the product of the bifunctional glucuronyltransferase and xylosyltransferase activities of LARGE), we tested each for elongation by B4GAT1 and LARGE. Incubation of the trisaccharides with UDP-GlcA and LARGE produced tetrasaccharide products in all cases, which were observed by HPLC and confirmed by LC-MS/MS (*Figure 5*). In sharp contrast, we were

**Table 1.** Kinetics at 2 mM UDP-GlcA

| Enzyme | Substrate | $K_m$ (mM) | $k_{cat}$ (s⁻¹) | $k_{cat}/K_m$ (M⁻¹ s⁻¹) | Specific activity (pmol/min/µg) |
|---|---|---|---|---|---|
| B3GNT1 | Xyl-α-pNP | 3.0 | 0.087 | 29 | 130 |
| | Xyl-β-pNP | 4.0 | 0.25 | 63 | 380 |
| LARGE | Xyl-α-pNP | 6.0 | 0.005 | 0.84 | 2.7 |
| | Xyl-β-pNP | NM | NM | NM | NM |

NM = not measurable (below limit of detection).

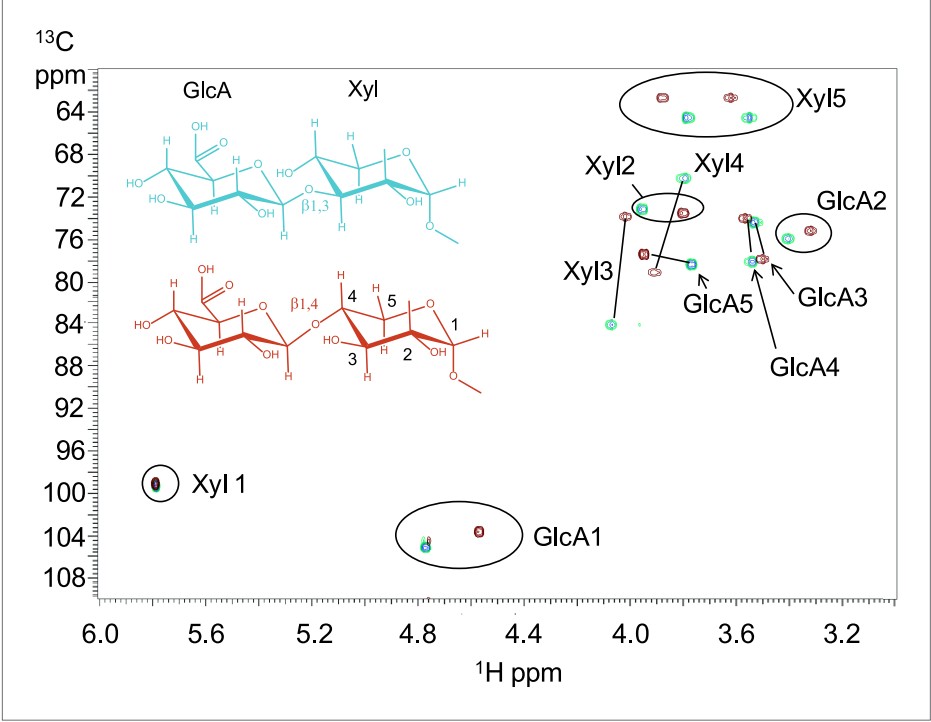

**Figure 3**. B3GNT1 is a β1,4-glucuronyltransferase in contrast to LARGE which is a β1,3-glucuronyltransferase. $^{1}H$–$^{13}C$ HSQC spectrum with peak assignments and structures of the products resulting from the action of B4GAT1 and UDP-GlcA on Xyl-α-pNP (red) and LARGE and UDP-GlcA on the same substrate (blue). Cross peaks are at the intersection of the $^{13}C$ and $^{1}H$ shifts of the partners in each C-H pair. The changes in the spectrum between the two compounds reflect the differences in the site of the glycosidic linkage formed, and particularly the major differences in the carbon shifts for the Xyl 3 and Xyl 4 sites between the two disaccharides is diagnostic of the change in their participation in the two different respective glycosidic linkages. The β configuration of the GlcA 1 site is confirmed by the 1-bond $^{1}H$–$^{13}C$ couplings of 166 Hz for these sites, as well as couplings between the H1 and H2 protons, 7.9–8.0 Hz. (data not shown).

The following figure supplement is available for figure 3:

**Figure supplement 1**. NMR determination of the 1,4 glycosidic linkage.

---

unable to produce any observable tetrasaccharide with B4GAT1 using any of the three trisaccharides as acceptors (*Figure 5*), though the enzyme was active for transfer to monosaccharide xylopyranosides (*Figure 2*).

## Neither B4GAT1 nor LARGE appear to possess branching activity

The inability of B4GAT1 to modify trisaccharides ending with xylose does not rule out the possibility that B4GAT1 may possess a branching activity. We tested whether B4GAT1 or LARGE were capable of transferring glucuronic acid to any of the tagged purified disaccharides GlcA-β4-Xyl-α-pNP, GlcA-β4-Xyl-β-pNP or GlcA-β3-Xyl-α-pNP. We did not detect any transfer of GlcA in these in vitro assays (*Figure 5—figure supplement 1*). Thus, none of the GlcA-capped products of the enzymes appear to be substrates for the other enzyme.

## Formation of disaccharide polymer lags in the absence of B4GAT1

To further characterize the interplay between LARGE and B4GAT1, we examined production of polymer by LARGE both with and without B4GAT1. Incubation of Xyl-α-pNP or Xyl-β-pNP with either B4GAT1 alone, LARGE alone, or B4GAT1 and LARGE with both sugar nucleotide donors resulted in substantially different polymer production time courses as assayed by radioactive transfer (*Figure 6*). Time points were taken at 1, 2, 14 and 36 hr. B4GAT1 rapidly completes transfer to substrate. ~1200 DPM represents complete transfer to the 10 nmol of substrate present. LARGE is unable to transfer to

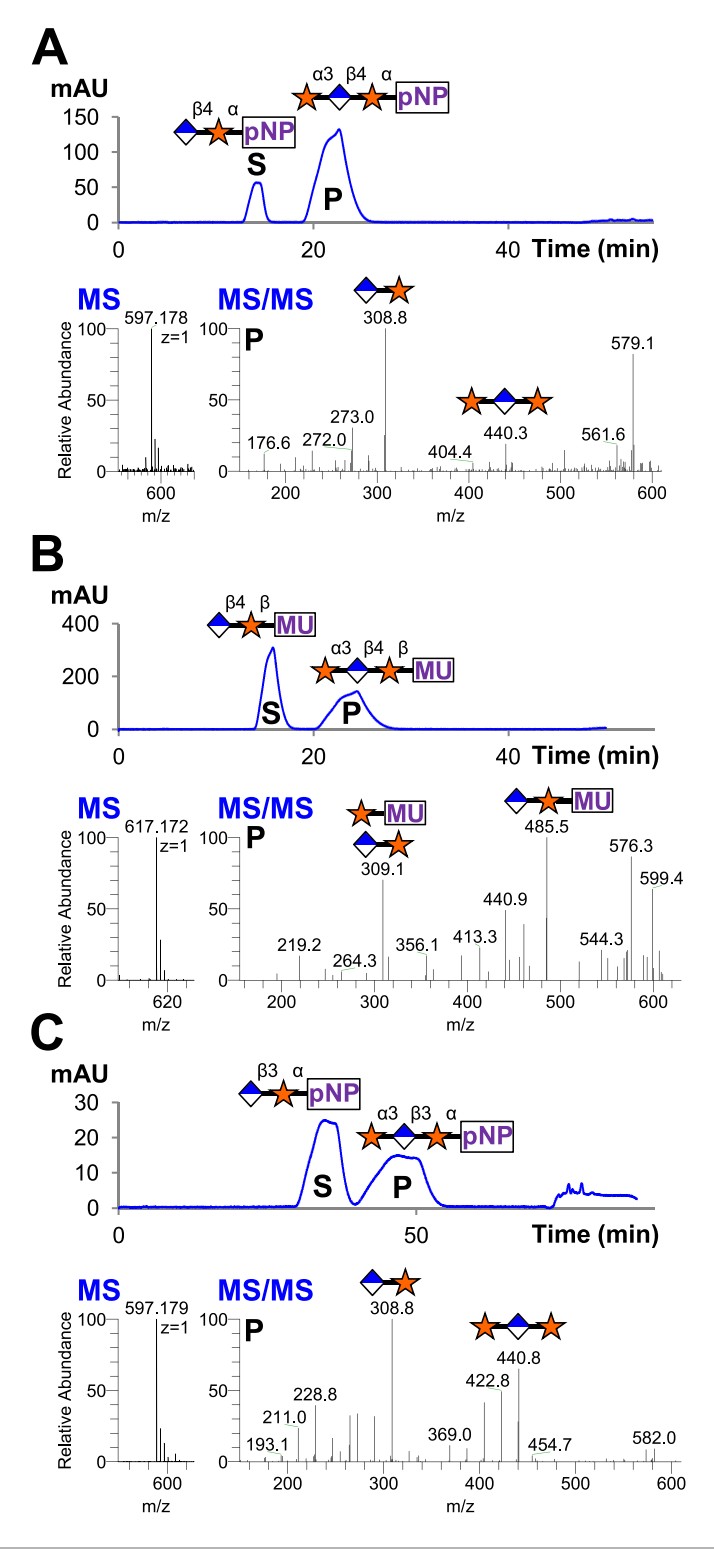

**Figure 4**. B4GAT1 disaccharide products are substrates for the xylosyltransferase activity of LARGE. S indicates substrate peaks, P indicates product peaks (xylose added). Separation was carried out by isocratic C18 reverse-phase HPLC with absorbance monitoring at 310 nm. Products were confirmed by accurate mass and MS/MS fragmentation shown below with assignment of key characteristic peaks. (**A**) P = Xyl-α3-GlcA-β4-Xyl-α-pNP. (**B**) P = Xyl-α3-GlcA-β4-Xyl-β-MU. (**C**) The product of the sequential dual enzymatic activity of LARGE as a positive

*Figure 4. Continued on next page*

*Figure 4. Continued*

control; P = Xyl-α3-GlcA-β3-Xyl-α-pNP. Reactions were carried out overnight in 0.1 M MES pH 6.5 containing 10 mM MnCl2, 5 mM MgCl2, 2 mM substrate and 2 mM UDP-Xyl.

Xyl-β-pNP (*Figure 6B*) and shows a significant lag in initial transfer to Xyl-α-pNP before the formation of polymer. In contrast, B4GAT1 and LARGE together demonstrate robust initial transfer activity and formation of polymer using either substrate. Analysis of the synthesized polymer via mass spectrometry shows extensive polymerization is achievable following incubation with both enzymes (*Figure 6—figure supplement 1*). These data provide further evidence that LARGE possesses poor initiating activity but works efficiently in polymer formation once B4GAT1 is added for initial transfer of GlcA.

## Discussion

O-mannosylation of α-dystroglycan is required for its proper function and when disrupted is a significant cause of a subset of congenital muscular dystrophies referred to as secondary dystroglycanopathies (*Barresi and Campbell, 2006*; *Freeze, 2007*; *Muntoni et al., 2011*; *Mercuri and Muntoni, 2012*; *Dobson et al., 2013*; *Wells, 2013*; *Praissman and Wells, 2014*). A phosphorylated O-mannose trisaccharide (core M3, [*Praissman and Wells, 2014*]) attached to α-dystroglycan and its extension by LARGE after poorly defined intermediate biosynthetic steps (*Figure 7*) has been shown to be directly involved in the required ECM ligand binding activity (*Yoshida-Moriguchi et al., 2010*; *Inamori et al., 2012*; *Yoshida-Moriguchi et al., 2013*). Here, we have established that B4GAT1 is a β-1,4-glucuronyltransferase, not a B3GNT (*Figure 1*), with activity toward monomeric α- and β- xylopyranosides (*Figure 2*, *Figure 3*). We have gone on to show that LARGE can extend the resulting products of B4GAT1 (*Figure 4*). Prior reports indicated that B4GAT1 (B3GNT1) activity is necessary for expression of LARGE-dependent functional glycosylation of α-dystroglycan and that B4GAT1 (B3GNT1) and LARGE form a complex (*Bao et al., 2009*; *Buysse et al., 2013*; *Czeschik et al., 2013*; *Shaheen et al., 2013*). We were unable to confirm this latter point using our soluble forms of the enzymes (*Figure 2—figure supplement 1*). Since we have shown that the specific activity and catalytic efficiency of B4GAT1 towards the tested monosaccharide xylosides is greater by more than an order of magnitude compared to LARGE catalysis (*Table 1*), this suggests that B4GAT1 is an initiating enzyme for LARGE-dependent glycan synthesis. This conclusion is further strengthened by data showing a significant lag in the initial rate for synthesis of or a complete failure to produce polymer in the absence of B4GAT1 (*Figure 6*). It also appears that B4GAT1 serves exclusively as a priming enzyme since we could not detect B4GAT1 modification of LARGE extended trisaccharides (*Figure 5*). We also demonstrate that neither LARGE nor B4GAT1 is capable of adding GlcA to a disaccharide that has been capped with GlcA suggesting that branching is not occurring (*Figure 5*, *Figure 5—figure supplement 1*). This leads to the conclusion that there is only one type of repeating disaccharide (Xyl-α3-GlcA-β3) attached to α-dystroglycan that is synthesized by LARGE. This data also serves to highlight the differences in structural isomers by showing both anomeric configuration (α and β, *Figures 2C and 6*) as well as linkage position (−3 and −4, *Figure 5*) of isomeric substrates influences their subsequent ability to be elongated by enzymes (LARGE and B4GAT1). In sum, our data provide a basis for further attempts to fully elucidate the functional LARGE-dependent O-mannose-initiated structure(s) and suggests that at least one of the remaining incompletely characterized genes implicated in functional glycosylation of α-dystroglycan (FKTN, FKRP, ISPD, and TMEM5, [*Jae et al., 2013*; *Wells, 2013*]) likely encodes a xylosyltransferase to provide a substrate for B4GAT1. Given that B4GAT1 is essential for proper glycosylation of α-dystroglycan (*Bao et al., 2009*; *Buysse et al., 2013*; *Czeschik et al., 2013*; *Shaheen et al., 2013*) and that LARGE is incapable of transferring GlcA to a β-xylopyranoside (*Figure 2*, [*Inamori et al., 2012, 2013*]) or generating a polymer from pNP-β-Xyl without B4GAT1 present (*Figure 6*), our findings strongly suggest that the underlying xylose is in a β-linkage on α-dystroglycan. This work also clarifies why deficiencies in this enzyme would be associated with loss of functional glycosylation and laminin binding of α-dystroglycan and be causal for congenital muscular dystrophy (*Bao et al., 2009*; *Buysse et al., 2013*; *Czeschik et al., 2013*; *Shaheen et al., 2013*). Further, our results likely clarify how loss of expression of B4GAT1 (B3GNT1), similar to loss of expression of LARGE (*de Bernabe et al., 2009*), can lead to loss of laminin-binding α-dystroglycan and promote metastasis in certain cancers (*Bao et al., 2009*).

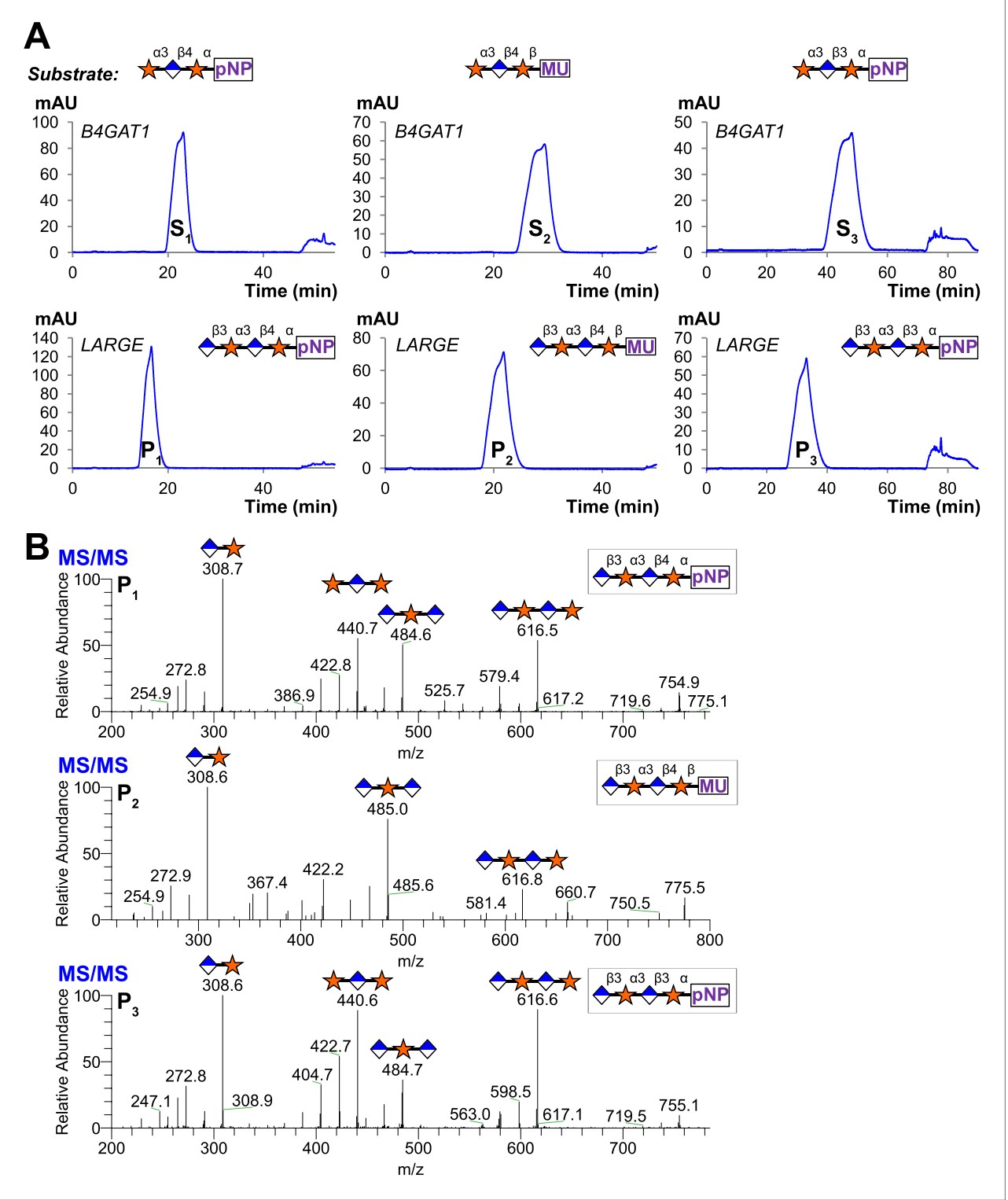

**Figure 5**. Trisaccharides terminating in an a3-xyloside are inefficient substrates for B4GAT1 in contrast to LARGE. Purified trisaccharides were incubated with LARGE or B4GAT1 plus UDP-GlcA overnight and then separated by isocratic reverse-phase C18 HPLC. (**A**) P with a subscript indicates product (addition of GlcA) whereas S with a subscript indicates an unmodified trisaccharide substrate. The top chromatogram in each pair shows the result of incubation with B4GAT1, the bottom shows the result of incubation with LARGE. (**B**) Results were confirmed by MS/MS fragmentation spectra, only product spectra are shown. Reactions were carried out overnight in 0.1 M MES pH 6.5 containing 10 mM MnCl2, 5 mM MgCl2, 2 mM substrate and 2 mM UDP-GlcA.

*Figure 5. Continued on next page*

*Figure 5. Continued*

The following figure supplement is available for figure 5:

**Figure supplement 1**. B4GAT1 and LARGE do not appear to possess branching activity.

Finally, potential complex interactions between poly-N-acetyllactosamine, as often measured by the i-antigen antibody, and the LARGE-dependent repeating disaccharide, often measured on α-dystroglycan by the IIH6 antibody, need to be reexamined in the context of the newly defined enzymatic activity of B4GAT1 (B3GNT1) (*Ujita et al., 2000*; *Bao et al., 2009*; *Lee et al., 2009*; *Wright et al., 2012*; *Yoneyama et al., 2012*; *Buysse et al., 2013*; *Czeschik et al., 2013*; *Shaheen et al., 2013*). For example, B4GAT1 has previously been observed to be in complex with B4GALT1, an enzyme required for poly-N-acetyllactosamine synthesis (*Lee et al., 2009*). This finding suggested that interaction between these two enzymes, which were thought to generate the repeating disaccharide, potentially influenced poly-N-acetyllactosamine synthesis. In lieu of our findings, this extrapolated conclusion from the interaction data is no longer valid. Instead, the relationship between poly-N-acetyllactosamine synthesis and the functional O-mannose glycan pathway needs to be evaluated, especially given that both have been implicated in cancer metastasis (*Ujita et al., 2000*; *Bao et al., 2009*; *Lee et al., 2009*; *Wright et al., 2012*; *Yoneyama et al., 2012*; *Buysse et al., 2013*; *Czeschik et al., 2013*; *Shaheen et al., 2013*).

We propose renaming B3GNT1 according to its defined activity as B4GAT1 (*Figure 1*, *Figure 2*, *Figure 3*). This designation is following in the tradition of the three other known non-proteoglycan glycoprotein glucuronyltransferases that all add GlcA in a β3-linkage (B3GAT1-3, [*Morita et al., 2008*]). Interestingly, the only established glycoprotein glycoslytransferases capable of transferring GlcA in a β4-linkage are the dual-activity exostoses enzymes involved in heparan sulfate (HS) proteoglycan

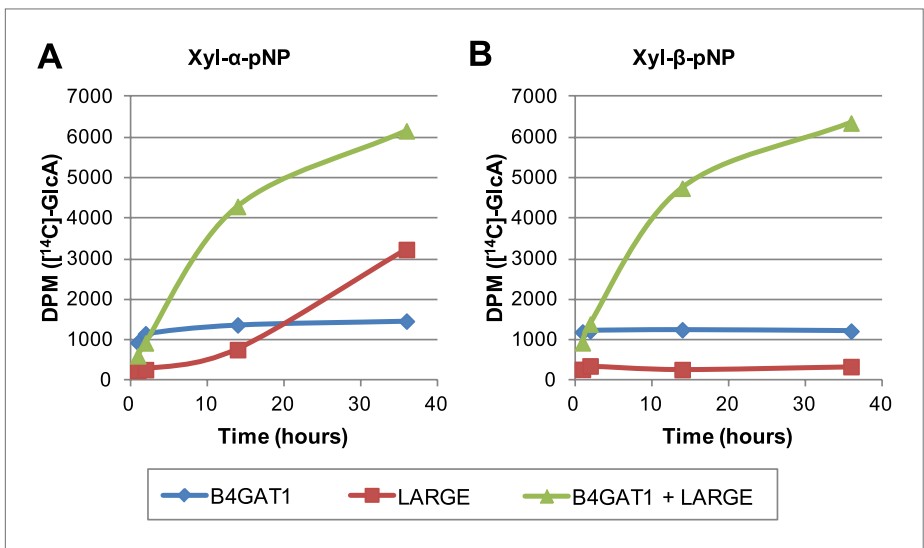

**Figure 6**. The slow reaction velocity of LARGE is rescued by addition of B4GAT1. Reactions were carried out in 0.1 M MES pH 6.5 with 10 mM MnCl2, 5 mM MgCl2, 40,000 DPM UDP-[¹⁴C]GlcA and 10 mM non-radioactive UDP-GlcA and UDP-Xyl each. Aliquots were removed at the displayed time points, boiled, and processed using RP C18 spin columns to separate untransferred donor from substrate. (**A**) The initial transfer of GlcA by LARGE to Xyl-α-pNP is slow in the absence of B4GAT1 but after transfer of the first GlcA polymerization rates increase to mirror those of LARGE in the presence of B4GAT1. (**B**) With Xyl-β-pNP as substrate, B4GAT1, which only possesses glucuronyltransferase activity, transfers a single GlcA per molecule of substrate. LARGE is unable to transfer GlcA to Xyl-β-pNP. Polymerization is only observed with the addition of both enzymes to the reaction mixture.

The following figure supplement is available for figure 6:

**Figure supplement 1**. B4GAT1 and LARGE combined produce extended GlcA-Xyl polymer chains.

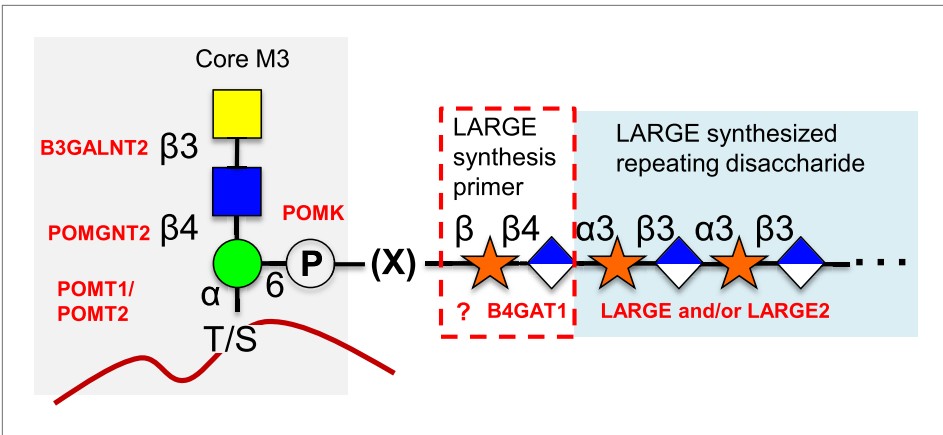

**Figure 7**. Proposed role of B4GAT1 in the O-mannosylation pathway. The molecular link between the phospho-O-mannose trisaccharide synthesized on α-dystroglycan and the LARGE synthesized repeating disaccharide crucial for laminin binding reactivity is still not fully characterized (represented as [X]). B4GAT1 appears to possess a priming activity for LARGE and likely adds to an underlying β-xylose that is added by an as yet undefined glycosyltransferase (represented with a question mark).

polymerization that build the repeating disaccharide (-GlcNAc-α4-GlcA-β4-) backbone of the glycosaminoglycan (*Zak et al., 2002*). Similarities exist between HS (and other proteoglycan) biosynthesis pathways and the LARGE-dependent functional O-mannose glycan assembly pathway built on α-dystroglycan (*Figure 7*). Included in these similarities are that they both contain non-reducing end repeating disaccharides (GlcA and GlcNAc for HS and GlcA and Xyl for the functional O-Man structure), they both contain an acidic glycan (GlcA), there is a copolymerase that has dual–enzymatic activity (exostoses enzymes for HS and LARGE for the functional O-Man glycan structure), both require a specific underlying core structure, and each contains specific priming glycosyltransferases (EXTLs for HS [*Kitagawa et al., 1999*; *Kim et al., 2001*], B4GAT1 for the functional O-Man glycan structure) that adds one of the sugars found in the repeat to the underlying core structure to initiate elongation. Here we have established that B4GAT1, previously referred to as B3GNT1, is a xylopyranoside β1,4-glucuronyltransferase that appears to be the priming enzyme for the LARGE copolymerase for building the functional O-Man structure on α-dystroglycan that when defective causes CMD.

## Materials and methods

### Reagents
Tagged monosaccharide glycosides were purchased from Sigma–Aldrich (St. Louis, MO) at ≥97% purity as were UDP-GlcA trisodium salt, UDP-GlcNAc disodium salt and UDP-Gal disodium salt. HPLC solvents were also purchased from Sigma–Aldrich (Chromasolv grade). UDP-Xyl was purchased from CarboSource Services (Athens, GA). Mini-protean TGX PAGE gels were purchased from Bio-Rad. UDP-[³H]GlcNAc was purchased from American Radiolabeled Chemicals (St. Louis, MO) and UDP-[¹⁴C]GlcA was from PerkinElmer (Waltham, MA).

### Enzyme production
The catalytic domains of human B3GNT1 (amino acid residues 54–415, UniProt O43505), B3GNT2 (amino acid residues 35–397, UniProt Q9NY97), and LARGE (amino acid residues 91–756, UniProt O95461) were expressed as soluble, secreted fusion proteins by transient transfection of HEK293 suspension cultures (*Meng et al., 2013*). The coding regions were amplified from Mammalian Gene Collection Gerhard, 2004 #4 clones using primers that appended a tobacco etch virus (TEV) protease cleavage site (*Phan et al., 2002*) to the NH₂-terminal end of the coding region and attL1 and attL2 Gateway adaptor sites to the 5′ and 3′ terminal ends of the amplimer products. The amplimers were recombined via BP clonase reaction into the pDONR221 vector and the DNA sequences were confirmed. The pDONR221 clones were then recombined via LR clonase reaction into a custom Gateway

adapted version of the pGEn2 mammalian expression vector (*Barb et al., 2012*; *Meng et al., 2013*) to assemble a recombinant coding region comprised of a 25 amino acid $NH_2$-terminal signal sequence from the *T. cruzi* lysosomal α-mannosidase (*Vandersall-Nairn et al., 1998*) followed by an 8xHis tag, 17 amino acid AviTag (*Beckett et al., 1999*), 'superfolder' GFP (*Pedelacq et al., 2006*), the nine amino acid sequence encoded by attB1 recombination site, followed by the TEV protease cleavage site and the respective glycosyltransferase catalytic domain coding region.

Suspension culture HEK293f cells (Life Technologies, Grand Island, NY) were transfected as previously described (*Meng et al., 2013*) and the culture supernatant was subjected to Ni-NTA superflow chromatography (Qiagen, Valencia, CA). Enzyme preparations eluted with 300 mM imidazole were concentrated to ~1 mg/ml using an ultrafiltration pressure cell membrane (Millipore, Billerica, MA) with a 10 kDa molecular weight cutoff.

## Enzymatic reactions

All reactions for *Figure 2* through six were performed in 0.1 M MES pH 6.5, 10 mM MnCl2, 5 mM $MgCl_2$. Reactions summarized in *Figure 1* were performed with omission of $MgCl_2$, conditions that more closely match those in the original literature (*Sasaki et al., 1997*; *Zhou et al., 1999*). Non-radioactive nucleotide sugar donors were included at 2 mM for analytical procedures excluding the polymer production assays in which both UDP-GlcA and UDP-Xyl were included at 10 mM. Nucleotide sugar donors were included at up to 8 mM for preparative scale production of material for NMR or purification for further reactions. Radioactive nucleotide sugar donors were included at approximately 40,000 DPM per sample. Substrate concentrations for analytical procedures were kept constant at 2 mM except in the kinetics assays. All incubations were carried out at 37°C. Incubation times for kinetics were set at 2 hr for B4GAT1 and 16 hr for LARGE based on time course curves used to ensure adequate transfer while maintaining the initial rate condition required. All other analytical reactions were performed for 16–18 hr while preparative reactions involving LARGE were carried out for upwards of 24 hr with occasional addition of enzyme due to the low specific activity of LARGE with respect to certain substrates.

## Enzyme assays and product purification

Enzymatic reactions were stopped by boiling for 5 min, acidified to 0.1% TFA, and glycoside acceptors were separated from sugar nucleotide using reverse-phase C18 spin (The Nest Group, Inc., Southborough, MA) columns. Transfer was determined by scintillation counting or HPLC with LC-MS/MS verification of species. For scintillation counting, a PerkinElmer (Waltham, MA) Tri-Carb 2910 TR liquid scintillation counter was used with ScintSafe Plus 50% scintillation cocktail under standard settings for the isotope in question. HPLC was carried out on an Agilent (Santa Clara, CA) 1100 LC system equipped with variable wavelength absorbance detector set for monitoring pNP and MU derivatives (310 nm). Quantification was by peak area. Buffer A was 50 mM ammonium formate pH 4.3 and buffer B was 20% buffer A in 80% acetonitrile for all assays discussed. Separations were carried out by isocratic elution using a Grace Vydac 218 TP C18 column (5 μm particle size, 2.1 mm × 150 mm). All disaccharide products were separated using 14%B. Trisaccharide and tetrasaccharide separations were noted to be most strongly influenced by the identity of the underlying labeled disaccharide. Hence, all longer chain products and substrates extending GlcA-β4-Xyl-α-pNP were separated at 8%B, all products and substrates extending GlcA-β4-Xyl-β-MU were separated using 10%B, and all products and substrates extending GlcA-β3-Xyl-α-pNP were separated at 5%B. LC-MS and LC-MS/MS of reaction products was performed in positive mode on either a Thermo Fisher (Waltham, MA) Orbitrap XL or a Thermo Fisher Orbitrap Fusion utilizing short linear gradients from 0.1% formic acid in water to 0.1% formic acid in 80% acetonitrile. Full MS was acquired in the Orbitrap for accurate mass determination while glycoside MS/MS fragmentation spectra were obtained in the linear ion trap and assigned manually.

## Shotgun proteomics and protein SDS-PAGE

Shotgun proteomics was performed on tryptic digests of purified enzyme samples according to our standard protocol on a Thermo Fisher Orbitrap XL (*Porterfield et al., 2014*). Data was searched in Proteome Discoverer 1.4 using Sequest HT with the percolator node set at a 1% peptide false-discovery rate and a recent human reference proteome from Uniprot to which common contaminant protein sequences had been added. SDS-PAGE was also carried out according to standard protocols using the

Bio-Rad (Hercules, CA) Precision Plus Protein Kaleidoscope molecular weight ladder. Gels were Coomassie Brilliant Blue G-250 stained and visualized.

## In vitro complex formation assay

Uncleaved LARGE and TEV protease cleaved B4GAT1 were combined at equimolar concentrations in 20 mM HEPES pH 7 with 150 mM NaCl and 10 mM imidazole. Final protein concentration was 0.5 mg/ml. After incubation at 4°C for 15 min, the mixture was applied to Novagen Ni-NTA resin, washed three times with 20 mM imidazole in PBS pH 7 and eluted with 250 mM imidazole in PBS pH 7. The input and collected fractions were analyzed by SDS-PAGE and silver staining (*Porterfield et al., 2014*).

## Enzyme kinetics

Time course experiments were carried out to ensure that the initial rate condition required for Michaelis–Menten kinetic analysis was met. A 2 hr incubation at 37°C was found to be acceptable for B4GAT1 whereas at 16 hr incubation period was judged suitable for LARGE. We compared the activity of B4GAT1 to that of LARGE while varying monosaccharide acceptor concentration. Concentrations ranging from 200 nM to 2 mM were tested to generate typical Michaelis–Menten curves for each enzyme against the substrates Xyl-α-pNP and Xyl-β-pNP. UDP-GlcA was supplied at 2 mM in all reactions. Reactions were terminated by boiling and processed as described above with quantification by HPLC. A Lineweaver-Burk plot was generated in Microsoft Excel and kinetic parameters determined from it. We were unable to detect the presence of product for LARGE at 2 mM donor and acceptor after 16 hr with Xyl-β-pNP.

## NMR

Samples of the products from action of LARGE and B4GAT1 with UDP-GlcA on Xyl-α-pNP were purified as described above, and, for each, ~1 mg was dissolved in 0.5 ml $D_2O$. NMR spectra of each of these samples were obtained on a VNMRS 600 MHz spectrometer with a triple resonance HCN cold probe. 2-Dimensional $^1H$ double quantum COSY, and TOCSY, and $^1H$-$^{13}C$ HSQC and HMBC spectra (*Van de Ven, 1995*) were recorded using standard pulse sequences in the Agilent VnmrJ 4.3 software and processed in that software. The $^1H$ and $^{13}C$ peak assignments were made based on analysis of these data using the single and multiple bond connectivities these experiments reveal. The glycosidic linkages were established based on $^1H$-$^{13}C$ HMBC correlations in both directions across the glycosidic linkages (*Figure 3—figure supplement 1*). Anomeric stereochemistry was determined from the 1-bond C-H coupling of the anomeric sites where a value of less than 170 Hz indicates a β-linkage and one of more than 170 Hz indicates an α-linkage (*Bock and Pedersen, 1974*). These were further confirmed from the $^3J_{H1,H2}$ couplings of the respective sugar residues. $^1H$ chemical shifts were calibrated relative to the HDO signal at 4.77 ppm at 25°C and the $^{13}C$ shifts were then determined by the indirect referencing approach from the proton reference frequency (*Wishart et al., 1995*).

## LARGE synthesized polymer analysis by mass spectrometry

Following a protocol developed to significantly enhance spectra acquired for highly acidic carbohydrates (*Jacobs and Dahlman, 2001*), Nafion 117 solution was applied to a Bruker MSP 96 ground steel target and allowed to dry. Samples of polymerization reaction mixtures were mixed 1:1 with 2,5-DHB resuspended in 50% acetonitrile at 20 mg/ml and spotted on the dried Nafion 117 membrane. Matrix-assisted laser desorption ionization with time-of-flight detection mass spectrometry spectra were acquired using a Bruker Microflex.

## Acknowledgements

We would like to thank all members of the Wells, Live, and Moremen laboratories for helpful discussions. We would like to thank Dr Debra Mohnen for radiolabeled sugar nucleotide and Dr Malcolm O'Neill for provision of Nafion 117 and use of a Bruker microflex. A manuscript detailing similar findings was co-submitted with this work from the laboratory of Dr Kevin Campbell. This work was supported in part by a grant from NIGMS/NIH (R01GM111939 to LW and DHL), a program project grant from NIGMS/NIH (P01GM107012, LW co-PI and GJB, PI), and technology resource grants from NIGMS/NIH (P41GM103490, LW and KWM co-PIs and P41GM103390, DHL and GJB co-PIs and KWM, PI).

## Additional information

### Funding

| Funder | Grant reference number | Author |
|---|---|---|
| National Institute of General Medical Sciences | R01GM111939 | David H Live, Lance Wells |
| National Institute of General Medical Sciences | P01GM107012 | Geert-Jan Boons, Lance Wells |
| National Institute of General Medical Sciences | P41GM103490 | Kelley W Moremen, Lance Wells |
| National Institute of General Medical Sciences | P41GM103390 | David H Live, Geert-Jan Boons, Kelley W Moremen |

The funders had no role in study design, data collection and interpretation, or the decision to submit the work for publication.

### Author contributions

JLP, Conception and design, Acquisition of data, Analysis and interpretation of data, Drafting or revising the article, Contributed unpublished essential data or reagents; DHL, Acquisition of data, Analysis and interpretation of data; SW, AR, ZSC, Acquisition of data, Contributed unpublished essential data or reagents; G-JB, Conception and design, Contributed unpublished essential data or reagents; KWM, Conception and design, Drafting or revising the article, Contributed unpublished essential data or reagents; LW, Conception and design, Analysis and interpretation of data, Drafting or revising the article

## Additional files

### Supplementary files

• Supplementary file 1. Top 5 protein search results in each purified enzyme sample LC-MS/MS was carried out on tryptic digests of purified enzyme samples. Results filtered to peptide 1% false-discovery rate are displayed from a Proteome Discoverer 1.4 Sequest HT search against the Uniprot human database along with total score and number of peptide spectral matches for each protein assignment (# PSMs).

• Supplementary file 2. Chemical shifts of disaccharides. Chemical shifts of the disaccharide portion of the products of LARGE and B4GAT1 addition of GlcA to Xyl-α-pNP. Proton shifts are referenced to the HDO signal, 4.77 ppm at 25°C relative to DSS, and $^{13}$C shifts relative to DSS at 0 ppm were then determined using indirect referencing to the proton standard.

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
