## [Decision Letter]

Thank you for sending your work entitled “B4GAT1 is the Priming Enzyme for the LARGE-dependent Functional Glycosylation of α-Dystroglycan” for consideration at *eLife*. Your article has been favorably evaluated by Vivek Malhotra (Senior editor) and 3 reviewers, one of whom, Suzanne Pfeffer, is a member of our Board of Reviewing Editors, while Pamela Stanley has also agreed to reveal her identity.

The Reviewing editor and the other reviewers discussed their comments before we reached this decision, and the Reviewing editor has assembled the following comments to help you prepare a revised submission.

This manuscript describes an exciting new development in our understanding of the synthesis of a functional glycan required for α-dystroglycan to interact with laminin and the ECM. The data strongly support the authors' conclusions that the enzyme formerly called B3GNT1 should be renamed B4GAT1. However, the following points and questions should be addressed by the authors:

1) LARGE has a very low specific activity in these assays. The data in this manuscript should be compared directly for consistency with previous data published by the Campbell and Bakker groups (3 papers) in terms of incubation times, UDP-sugar concentrations, pH, metal ions etc. In addition, the precise conditions for each set of assay conditions should be given in all figure and table legends. It is not sufficient to have general conditions in the Methods.

2) It is intriguing that B4GAT1 has similar specificity for Xyl-α-MU and Xyl-β-MU. Could B4GAT1 be a branching enzyme as well as a priming enzyme? The fact that branching is not observed with tri-saccharide substrates under certain conditions *in vitro* does not rule out this possibility.

3) Does LARGE produce at least a small polymer from di- or trisaccharide substrate when both UDP-xylose and UDP-GlcA are present in the reaction? It would support priming if polymer is only obtained when B4GAT1 is in the mix.

4) The literature indicates that LARGE and B4GAT1 should form a complex. Is this the case for the purified enzymes? And, if so, does the complex increase the specific activity of LARGE/B4GAT1 for Xyl-α-MU? These data would provide important support for the priming activity of B4GAT1.

5) The study uses artificial substrates and *in vitro* assays. The model presented predicts that the alpha-dystroglycan from B4GAT1 deficient cells (KD, KO or patient derived) should display a single branched Xylose (or a terminal Xylose) in addition to the phospho-O-Mannose tri-saccharide. In recognition of the difficulty of obtaining sufficient quantities for structural evidence, indirect evidence that the absence of B4GAT1 results in the production of O-Mannose glycans on alpha-dystroglycan terminating in beta-xylose (or at least xylose) would be helpful.

6) To explore why B4GAT1 is indispensable and prove it is a priming enzyme, it would be ideal to test whether the first Xylose on the alpha-dystroglycan is indeed branched in Beta as proposed by the authors. Please speculate on how this could be tested.

7) Lee at al 2009 presents data for intracellular association of B4GAT1/B3GNT1 with B4GALT1. Perhaps the new findings suggest that a variety of Golgi glycosyltransferases are present in larger, oligomeric assemblies in cells? Please speculate. Also, could the authors speculate why the previous work came to the wrong conclusion?

---

## [Author Response]

*1) LARGE has a very low specific activity in these assays. The data in this manuscript should be compared directly for consistency with previous data published by the Campbell and Bakker groups (3 papers) in terms of incubation times, UDP-sugar concentrations, pH, metal ions etc. In addition, the precise conditions for each set of assay conditions should be given in all figure and table legends. It is not sufficient to have general conditions in the Methods*.

We fully agree that LARGE has a low specific activity in these assays using monosaccharides as substrates. The buffering conditions between our assays and those used by others (Inamori K, et al., 2012, Science, 335:93, Goddeeris, MM et al., 2013, Nature, 503:136, and Ashikov, A, et al., 2013, Glycobiology, 23:303, Inamori K, et al., 2013, Glycobiology, 23:295) are essentially the same. However, specific activity was only reported in the Inamori 2013 manuscript as the other experiments used undefined amounts of enzyme and rarely look at initial rates. The Inamori, et al. 2013 manuscript used a disaccharide for establishing rates due to the low activity observed for monosaccharide as a substrate consistent with our findings. As can be clearly observed in Figure 4, Figure 6, and Figure 6—figure supplement 1 of our manuscript, it is the initial transfer of GlcA to a monomer that appears to be slow and then extension appears to be robustly consistent with the observations of Inamori et al. 2013. Thus, we believe our results are fully consistent with what others have observed though direct comparison is not possible due to undefined reactions and/or different substrates being utilized for establishing kinetic parameters.

*2) It is intriguing that B4GAT1 has similar specificity for Xyl-α-MU and Xyl-β-MU. Could B4GAT1 be a branching enzyme as well as a priming enzyme? The fact that branching is not observed with tri-saccharide substrates under certain conditions in vitro does not rule out this possibility*.

Given the fact that the B4GAT1 and the LARGE glycosytransferase add GlcA in different linkages to α-xylose, this is an intriguing possibility. To better address this issue, we produced and purified the disaccharides GlcA-β3-Xyl-α-pNP (product of LARGE) and GlcA-β4-Xyl-α-pNP (product of B4GAT1). We performed a radioactive GlcA transferase activity with each of these substrates using both enzymes, as well as positive controls for activity (Figure 5—figure supplement 1). In our hands, we were unable to see transfer of radioactivity that would have been indicative for the formation of GlcA-β3-(GlcA-β4)-Xyl-α-pNP. While we are unable to completely rule out this possibility, since it is possible we are not using the correct substrates, our results argue against a branching activity for B4GAT1 or LARGE GlcA activity.

*3) Does LARGE produce at least a small polymer from di- or trisaccharide substrate when both UDP-xylose and UDP-GlcA are present in the reaction? It would support priming if polymer is only obtained when B4GAT1 is in the mix*.

The production of polymer with LARGE alone has been previously described by others using disaccharide substrates (Inamori K, et al., 2012, Science, 335:93, Goddeeris, MM et al., 2013, Nature, 503:136, and Ashikov, A, et al., 2013, Glycobiology, 23:303). To further address the issue of priming and polymer formation, we have included additional experiments (Figure 6 and Figure 6—figure supplement 1). This new data clearly demonstrates that LARGE alone is not able to form polymer using the Xyl-β-pNP as substrate (Figure 6). However, LARGE in the presence of both sugar nucleotides can produce polymer on Xyl-α- pNP though the initial rate, for the addition of the first GlcA, is quite slow (Figure 6). Further, the addition of B4GAT1 eliminates this initial lag observed with Xyl- α-pNP as the substrate (Figure 6). Using the Xyl-β-pNP, polymer is only observed when both enzymes are present (Figure 6). Furthermore, we confirmed the presence of polymer formed when both enzymes are used by mass spectrometry (Figure 6—figure supplement 1). This data strongly supports the model of B4GAT1 being the priming enzyme for LARGE-dependent polymer formation. Further, when coupled with the similar phenotype observed in patients and animal models with mutations/knock-down of B4GAT1 or LARGE, this data suggests that the underlying xylose on the alpha-dystroglycan structure is most likely in a beta-linkage.

*4) The literature indicates that LARGE and B4GAT1 should form a complex. Is this the case for the purified enzymes? And, if so, does the complex increase the specific activity of LARGE/B4GAT1 for Xyl-α-MU? These data would provide important support for the priming activity of B4GAT1*.

Tests carried out with our purified soluble enzymes did not reveal *in vitro* complex formation (Figure 2—figure supplement 1). This could potentially be attributed to the lack of transmembrane and stalk regions in our proteins. However, in a similar finding to ours, while Bao et al had previously demonstrated that B4GAT1 and LARGE when overexpressed as tagged proteins in cells together could co-immunopurify, they demonstrated that mixing of two cell extracts each overexpressing one of the proteins did not result in complex formation (Bao, X et al, 2009, PNAS, 106:12109). This suggests an unknown need for co-expression for complex formation or it reflects the inherent difficulty in determining specific protein-protein interactions via co-immunoprecipiation for membrane proteins.

While we were unable to see a complex, we were able to see an enhancement in initial rates for polymer formation when B4GAT1 was added with LARGE to Xyl- α-pNP and a complete inability to form polymer except when B4GAT1 was added with LARGE using Xyl-β-pNP as the substrate (Figure 6). This strongly supports the proposed priming activity of B4GAT1.

*5) The study uses artificial substrates and in vitro assays. The model presented predicts that the alpha-dystroglycan from B4GAT1 deficient cells (KD, KO or patient derived) should display a single branched Xylose (or a terminal Xylose) in addition to the phospho-O-Mannose tri-saccharide. In recognition of the difficulty of obtaining sufficient quantities for structural evidence, indirect evidence that the absence of B4GAT1 results in the production of O-Mannose glycans on alpha-dystroglycan terminating in beta-xylose (or at least xylose) would be helpful*.

We agree with reviewers and would direct them to the co-submitted article by Campbell’s laboratory that has the B4GAT1 deficient cells and determines the Xyl is terminal on alpha-dystroglycan in these cells and in a beta linkage based on glycosidase specificity. Our new data presented in Figure 6 that LARGE cannot generate polymer in the absence of B4GAT1 on Xyl-β-pNP is consistent with this argument as well.

*6) To explore why B4GAT1 is indispensable and prove it is a priming enzyme, it would be ideal to test whether the first Xylose on the alpha-dystroglycan is indeed branched in Beta as proposed by the authors. Please speculate on how this could be tested*.

See answer to question 5 above. Briefly, we have now demonstrated that LARGE cannot produce polymer in the absence of B4GAT1 on Xyl-β-pNP but can, albeit with a slow initial rate, on Xyl-α-pNP.

*7) Lee at al 2009 presents data for intracellular association of B4GAT1/B3GNT1 with B4GALT1. Perhaps the new findings suggest that a variety of Golgi glycosyltransferases are present in larger, oligomeric assemblies in cells? Please speculate*. *Also, could the authors speculate why the previous work came to the wrong conclusion?*

Given the often-times high Km’s of glycoslytransferases, the presence of oligomeric assemblies of enzymes in the secretory pathway would be a reasonable proposed mechanism for directed complex glycosylation. Of course, the oligosaccharyl transferase is a known multi-protein complex in the ER responsible for en bloc transfer of glycans from lipid linked oligosaccharides to Asn residues on proteins serving as a proof-of-concept for this model. Furthermore, a recent publication (Babu et al, 2012, Nature, 489:585) clearly established that there are multiple membrane protein-protein interactions occurring in eukaryotic cells. Lee et al 2009 established that B4GAT1 (now known to be involved in the LARGEdependent O-Mannosylation pathway [demonstrated here and in the companion article from the Campbell laboratory]) interacted with B4GALT1 (an enzyme involved in the generation of multiple types of glycans including, importantly in this context, the synthesis of polyLacNAc), and that they could influence each others location in the secretory pathway. We think it is overly strong to state that these authors arrived at a wrong conclusion as this manuscript’s major conclusion, in our view, was that these two enzymes interact. This appears to be a valid conclusion based on their studies that showed co-immunoprecipitation, co-localization, and the ability of one enzyme to regulate the localization of the other enzyme in the secretory pathway. At the time of these studies, one could postulate that these 2 enzymes associated together to more efficiently produce poly-LacNAc as the 2 activities needed are B3GNT and B4GALT activity. This of course needs to be re-evaluated in light of our findings. Being highly speculative, this complex could be involved in generating the phenomena seen in some cancer types where production of poly-LacNAc goes up and production of the LARGE-dependent glycosylation goes down. However, this remains to be evaluated.